# Effect of active follow-up of women with previous cesarean delivery on uptake of timely safe obstetric and surgical care: Comparison between pre-intervention and intervention cohorts in Rwanda

Josée Uwamariya[1]*, Gloriose Abayisenga[1], Ntwali Ndizeye[1], Fidele Nkurunziza[1], Albert Bisore Ngemanyi, Christian Mazimpaka[2], Jean Maurice Munyabarenzi[1], Rosine Bigirimana[1], Gilbert Rukundo[3], François Regis Cyiza[3], Farhad Khan[4], Renae Stafford[4], Karen Levin[4], Anne Fitzgerald Vinluan[5], Kathryn Mimno[5], Dieudonne Ndatimana[2]

1 USAID MOMENTUM Safe Surgery in Family Planning and Obstetrics/IntraHealth International, Kigali, Rwanda, 2 USAID Ingobyi Activity/IntraHealth International, Kigali, Rwanda, 3 Rwanda Biomedical Center, Kigali, Rwanda, 4 EngenderHealth, Washington, United States of America, 5 IntraHealth International, Chapel Hill, North Carolina, United States of America

* joseeuwa@yahoo.com

## Abstract

### Introduction

The rate of cesarean delivery (CD) in Rwanda has increased significantly from 2.2% in 2000 to 15.6% in 2020. Given increasing risks in subsequent pregnancy following CD it is important that women who have had a prior cesarean section plan and schedule CD in later pregnancies. This study assessed the effectiveness of the USAID MOMENTUM Safe Surgery in Family Planning and Obstetrics intervention in reducing emergency CDs among women with previous CDs.

### Methods

We conducted a cohort study in four public hospitals and 64 health centers across Rwanda, comparing two non-parallel cohorts: a pre-intervention cohort (December 2021–February 2022) and an intervention cohort (November 2022–May 2023). Exploratory data analysis and logistic regression were conducted to analyze the emergency CD rate and any associated factors.

### Results

The pre-intervention group comprised 212 women, whereas the intervention group involved 283 women, of whom 189 were included in the analysis. Among the 189 women in the intervention group, 87.3% reported to the hospital within five days post-referral when they were first called. The percentage of women who consulted

**Data availability statement:** We have uploaded the dataset to Figshare, where it is publicly available for access and reference. You can view and download the dataset a https://figshare.com/articles/dataset/Follow_up_of_pregnant_women_with_previous_cesarean_delivery_in_Rwanda_dataset_csv_csv/28341953?-file=52116479.

**Funding:** This study was funded by the United States Agency for International Development (USAID) through the MOMENTUM Safe Surgery in Family Planning and Obstetrics project, led by EngenderHealth and implemented in Rwanda by IntraHealth International. The views and opinions expressed in this paper are those of the authors and do not necessarily reflect those of USAID.

**Competing interests:** The authors have declared that no competing interests exist.

for a delivery plan within 36–38 weeks of gestation increased from 37.6% in the pre-intervention group to 62.4% in the intervention group. Consequently, in the adjusted logistic regression model, there was still a significant association between the intervention and reduced odds of emergency CD, with a 81% reduction in the odds of delivery by emergency CD (0.187; 95% CI: [0.115; 0.298]) compared to pre-intervention.

## Conclusion

This study demonstrates the effectiveness of an active follow-up intervention in promoting delivery planning and reducing emergency CD rates among pregnant women with previous CD scars. The comprehensive intervention, including tailored education and personalized phone conversations around the delivery period, appears to have contributed to increased awareness and motivation for women to seek timely care at the hospital for delivery planning.

## Introduction

Globally, the rate of cesarean deliveries (CDs) has been on the rise particularly in low- and middle-income countries (LMICs), with several countries experiencing several-fold increases in the population-level CD rate over the past 20 years [1]. In LMICs, CDs are often performed in response to complications, leading to a high prevalence of emergency CDs [2]. Sub-Saharan Africa, in particular, has a unique challenge, with health systems struggling to keep pace with the increasing demand for CDs while ensuring safety and efficiency [3].

The rate of CD in Rwanda has risen substantially, from 2.2% in 2000 to 15.6% in 2020 [4]. A study conducted in 2019 in rural areas of Rwanda identified a history of previous CDs as the primary indication for current CDs. This study also reported that an alarming 85.4% of these women presented at hospitals as emergency CD cases [5]. The pattern of high emergency CD prevalence is not isolated to Rwanda. This trend reflects broader patterns observed across comparable settings. For instance, a study at Uganda's Mulago Hospital reported that emergency CDs accounted for 85% of all CDs performed [6]. Given these statistics, pregnancies following a CD are considered high-risk and necessitate rigorous follow-up and adept management by specialized health care providers to optimize maternal and neonatal outcomes [7]. While emergency CD is a critical part of comprehensive emergency obstetric care, reducing the burden of avoidable emergency procedures can help to improve outcomes [8]. Further, in resource-limited settings like Rwanda, delivery planning for high risk cases can help to ensure that the necessary capabilities are available not only for these planned cesarean deliveries but also for managing other obstetric complications that may arise. Safe trial of labor after cesarean delivery (TOLAC) requires appropriate patient selection, continuous monitoring, and back-up advanced surgical services [9], which are mostly available in tertiary hospitals in Rwanda.

Recognizing the urgent need to address this issue, MOMENTUM Safe Surgery in Family Planning and Obstetrics, a US Agency for International Development (USAID)-funded project led by EngenderHealth and implemented in Rwanda by IntraHealth International, launched an active follow-up intervention targeting women with previous CD in Rwanda. The objective of this study is twofold: [1] to evaluate the effect of the intervention on the emergency CD rate among women with previous CD, and [2] to identify factors associated with emergency CD. The results of this study will not only contribute to an understanding of the intervention's effectiveness, but also provide valuable insights to guide policy and practice in managing post-CD pregnancies.

## Methodology

### Study design

This study compared emergency CD rates among a cohort of women with prior CD before the intervention was introduced with rates among a cohort of women with prior CD after the intervention was introduced. The cohort study design enabled the study team to examine the effects of the intervention by comparing outcome, defined as emergency CD rate before and during the intervention. In this study, a case was classified as an emergency CD if it met one of the following criteria a) a woman with a previous CD was admitted with contractions and a physician's documentation of the need for emergency intervention, or b) a woman with a previous CD that had a delivery plan and presented to the hospital on the scheduled date and was admitted without contractions; however, while awaiting the procedure, she began experiencing contractions and the physician confirmed the need for emergency intervention.

These criteria were applied consistently across both the pre-intervention and intervention cohorts to ensure uniformity in outcome classification.

### Setting

We implemented the study in four public hospitals (Kibagabaga, Nyamata, Nyanza, and Ruhengeri) and 64 health centers that refer patients to these hospitals. The selected hospitals and health centers receive technical and operational support provided by IntraHealth International through the MOMENTUM Safe Surgery project. We selected the hospitals for the study based on their high annual rate of CDs, which were the highest among the 26 hospitals supported by the project. The selected hospitals are located in urban areas but serve as referral centers for surrounding health centers in both urban and rural areas.

### Facility characteristics during the study period

Kibagabaga district hospital (DH) is staffed with two gynecologists, 40 midwives, and 15 anesthetists, with staffing levels remaining consistent across both the pre-intervention and intervention periods. The average monthly delivery rate was 570 deliveries during the pre-intervention period and 507 deliveries during the intervention period. The hospital serves as a referral center for patients transferred from 16 surrounding health facilities within Gasabo district. The average annual number of deliveries within the hospital's catchment area is 24,765, with a CD rate of 22.7%.

Ruhengeri referral hospital (RH) employs two gynecologists, 23 midwives, and seven anesthetists with staffing levels remaining consistent across both the pre-intervention and intervention periods. The average monthly delivery rate was 499 deliveries during the pre-intervention period and 562 deliveries during the intervention period. The hospital receives patients transferred from 16 surrounding health centers within Musanze district. The average annual number of deliveries within the hospital's catchment area is 13,837, and the CD rate is 23.9%.

Nyanza DH has one gynecologist, 21 midwives, and six anesthetists with staffing levels remaining consistent across both the pre-intervention and intervention periods. The average monthly delivery rate was 373 deliveries during the pre-intervention period and 378 deliveries during the intervention period. The hospital serves as a referral center for

patients transferred from 17 surrounding health facilities within Nyanza district. The average annual number of deliveries within the hospital's catchment area is 8,945, with a CD rate of 26.3%.

Nyamata DH employs two gynecologists, 25 midwives, and six anesthetists with staffing levels remaining consistent across both the pre-intervention and intervention periods. The average monthly delivery rate was 499 deliveries during the pre-intervention period and 491 deliveries during the intervention period. The hospital receives patients transferred from 15 surrounding health facilities within Nyamata district. The average annual number of deliveries within the hospital's catchment area is 15,351, and the CD rate is 17.7%.

## Intervention description

The active follow-up of pregnant women with previous CD is implemented at the health centers' antenatal care (ANC) units and begins when a woman with a medical history of previous CD visits the ANC service unit (Fig 1). The intervention is multipronged and utilizes education and phone call reminders to encourage women with a previous CD to seek timely obstetric care. The intervention reinforces education focusing on the risks related to TOLAC without adequate support, planned repeat CD, and the importance of seeking timely care to achieve a safe birth and avoid the risks of repeat emergency CD. The education sessions continue until the last ANC visit at 36 + weeks of pregnancy when the women receive a

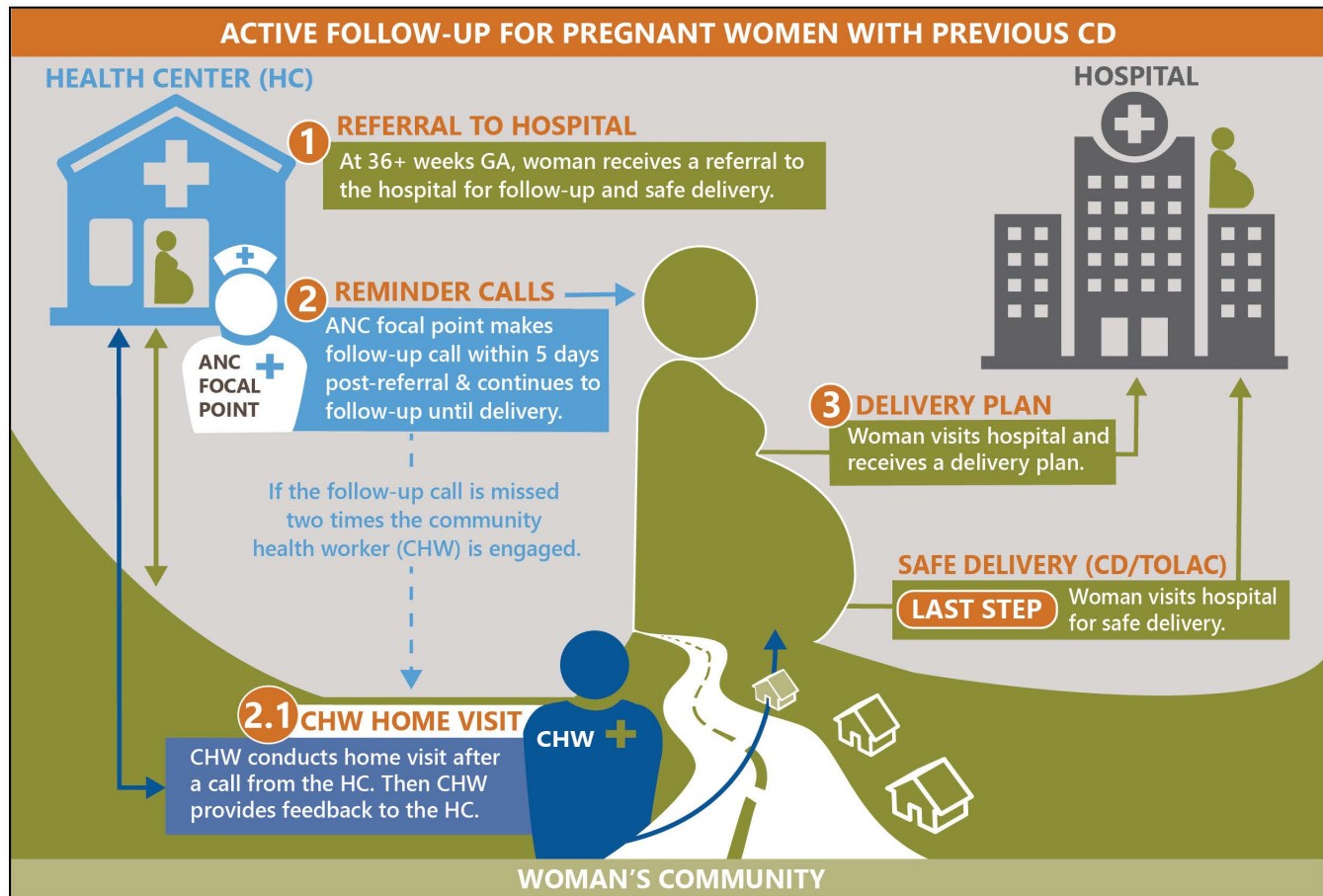

**Fig 1. Schematic of active follow-up intervention.**

referral to the hospital for delivery planning (step 1): an obstetric checkup where expectant mothers and health care providers at the hospital engage in discussions to explore safe options for childbirth and schedule a date for delivery. In addition to education, women are informed that they will receive a phone call from healthcare providers shortly after referral.

Phone call reminders are initiated after referral to ensure that the referred women report to the hospital, receive a birth plan including a delivery date, and adhere to it to avoid an emergency CD (step 2). Typically, the first call is made within five days of the referral to confirm if the woman had visited the hospital and received a birth plan and delivery date (step 3). If the woman has not gone to the hospital, the ANC nurse asks her when she plans to go and schedules the next reminder accordingly. The ANC nurse sends the hospital a list of all referred women on a weekly basis and uses this opportunity to get feedback on previously referred women.

In cases where the ANC nurse cannot reach the referred women, community health workers (CHWs) are engaged (step 2.1). The ANC nurse calls a CHW for a home visit to understand why the referred woman has not visited the hospital and to provide counseling and encouragement. The CHW follows up closely until the woman reports to the hospital and reports back to the health center. The phone call reminders are closed when information has been received that a referred woman has been admitted to the hospital for delivery or has moved or delivered in another setting outside the hospital's catchment area.

## Study population

The pre-intervention cohort included women with a history of previous CD who delivered at the study hospitals from December 2021 to February 2022. The intervention cohort consisted of pregnant women who had a history of previous CD and received the active follow-up intervention from ANC nurses, and who delivered at the study hospitals from November 2022 to May 2023. The pre-intervention group comprised 212 women, whereas the intervention group involved 283 women.

## Data source

Key variables of interest, including demographic information, characteristics of women before and during pregnancy, ultrasound examination, pregnancy outcomes, and newborn outcomes, were extracted from various sources such as ANC registers, electronic medical records systems, delivery registers, and patients' files. Patients' identifiable information was not collected.

## Data collection and analysis

The study employed a pre-intervention and intervention cohort design to compare outcomes, focusing on the emergency CD rate before and during the intervention period. The intervention involved active follow-up of pregnant women with previous CD scars through education and phone call reminders. Data collection for both periods was retrospective and conducted by trained data collectors from February 2023 to June 2023 using a structured Kobo Toolbox questionnaire. Data collection for the intervention period focused on participants who received the intervention and gave birth between November 2022 and May 2023. In contrast, the pre-intervention period focused on participants who did not receive the intervention and gave birth between November 2021 and February 2022. Missing values in outcomes were excluded from the analysis, while efforts were made to retrieve missing data for independent variables from health facility registers.

We conducted exploratory data analysis to examine the association between the mode of delivery, study period, and participants' characteristics. This analysis included calculating frequencies, percentages, and percent changes. Additionally, bivariate and multivariable logistic regression models were used to assess the factors associated with emergency delivery for unadjusted and adjusted factors respectively. All the analyses were conducted using Stata V.17 software.

## Sample size calculation

The determination of sample size needed for each study period relied on formulae designed to calculate the requisite sample size for comparing two groups [10].

$$\text{Sample size} = \left(2 * \left(Z_{\left(\frac{\alpha}{2}\right)} + Z_\beta\right)^2 * P(1-P)\right)/([P_1 - P_2)]^2),$$

where:

$$Z_{(\alpha/2)} = Z_{(0.05/2)} = 1.96 \text{ from Z table at type I error of 5\%}$$

$$Z_\beta = Z_{0.20} = 0.842 \text{ from Z table at 80\% power}$$

P1-P2 = Difference in proportion of event from the two study periods. From a study done previously at Kirehe DH, the proportion of women giving birth via an emergency CD following a previous CD pregnancy was 85.4% [11]. From this study we expect that a decrease of at least 12% from 85.4% to 73.4% would be considered significant.

$$P = \text{Pooled prevalence } (0.854 + 0.734)/2 = 0.794$$

$$\text{Sample size} = (2 * (1.96 + 0.842)^2 * 0.794(1 - 0.794))/([0.854 - 0.794])^2) = 178,$$

The minimum calculated sample size for each study period is 178 participants. To accommodate potential missing values for the outcome of interest, this sample size was increased by 7%, resulting in a total sample size of 191 participants for each study period.

## Ethical consideration

The study received approval from the IntraHealth International Institutional Review Committee (NO.21010) and the Rwanda National Ethics Committee (NO. 48/RNEC/2022). Since this study relied on retrospective data sources, the requirements for written informed consent were waived.

## Results

In this study, we enrolled 283 women with a history of previous CD for the intervention period and compared them with 212 women whose data were obtained from electronic medical records of participating health facilities. Among the women in the intervention group, 78 had missing outcomes and 16 delivered vaginally, leading to their exclusion from the analysis. Consequently, only 189 women were included in the analysis for the intervention period, while 212 women were considered for the pre-intervention period (Fig 2).

The study included a total of 401 women, among whom the majority (67.3%) were aged between 25 and 35 years. The analysis of women's demographics revealed no significant differences except for the health facilities where the women were enrolled. Specifically, during the intervention period, the number of participants from Nyamata DH (10.6%) and Ruhengeri RH (16.4%) was lower compared to the pre-intervention period. Additionally, among the 189 women in the intervention group, 87.3% had reported to the hospital within five days post-referral when they were first called: 66.7% consulted with the hospital for birth planning and were scheduled for delivery, and 20.6% were immediately admitted for delivery. Furthermore, the percentage of women who had developed a delivery plan by 36–38 weeks of gestation

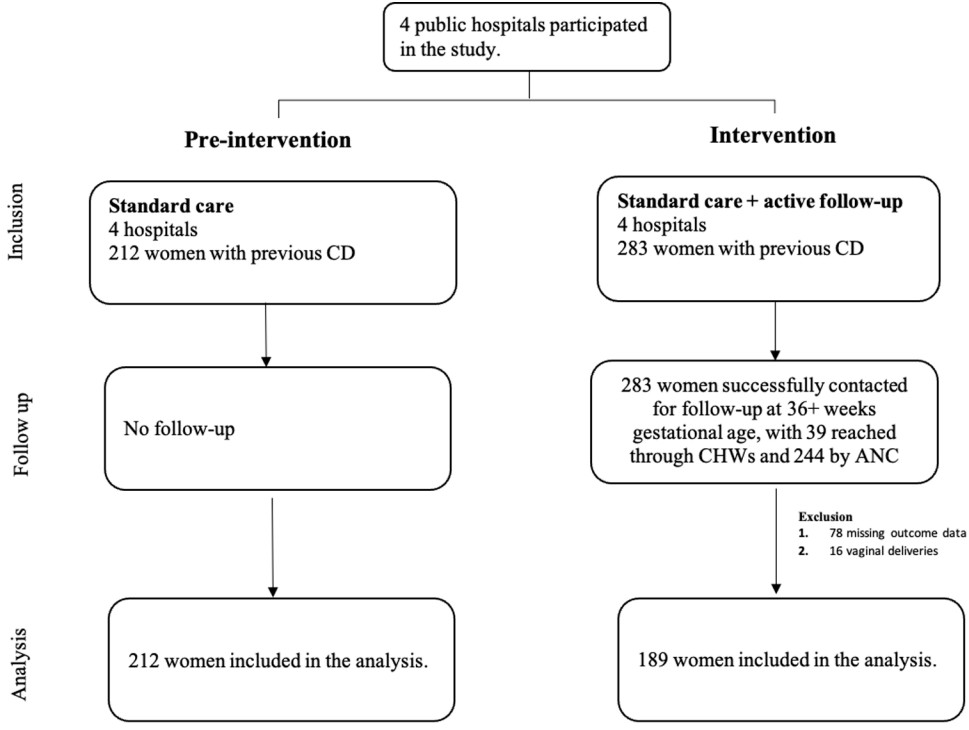

**Fig 2. Procedure for selecting study participants.**

increased from 37.6% in the pre-intervention group to 68.6% in the intervention group. Consequently, the percentage of women reaching the hospital post-referral after 38 weeks of gestation decreased by 50%, from 62.4% to 31.4%. Correspondingly, there was a notable decrease in the percentage of women presenting at referral health facilities with uterine contractions, declining from 62.3% in the pre-intervention period to 22.2% during the intervention phase. (Table 1).

The analysis of facility distribution revealed variations in CD rates, with Kibagabaga DH exhibiting the lowest proportion of emergency CD (45.9%). Significant associations were observed between the performance of ultrasound examinations during pregnancy and the incidence of emergency CD, with women receiving ultrasound examinations demonstrating a notably lower percentage of emergency CD (43.9%) compared to those who did not undergo such examinations (69.0%). Furthermore, preterm deliveries (≤36 weeks) were associated with a higher likelihood of emergency CD (87.0%) compared to term deliveries (>36 weeks). Moreover, women presenting with uterine contractions upon arrival at the hospital had a significantly higher incidence of emergency CD (90.1%) compared to those without such contractions (21.4%). Furthermore, the intervention period demonstrated a notable decrease in the percentage of emergency CD (31.7%) compared to the pre-intervention period (69.3%) (Table 2).

The logistic regression analysis shed light on the factors influencing the occurrence of planned and emergency CD (Table 3). In the unadjusted model, the intervention was significantly associated with a 79% reduction in the odds of delivering by emergency CD (odds ratio: 0.21; 95% CI: 0.13–0.31) compared to the pre-intervention period. However, after adjusting for confounding variables such as maternal age, hospital, performance of ultrasound examinations during pregnancy, the adjusted model showed that the odds of undergoing an emergency CD were reduced by 81% (0.19; 95% CI: [0.115; 0.298]) for women in the intervention group compared to the pre-intervention period. Additionally, Women who received an ultrasound examination at the hospital during their pregnancy had significantly lower odds of requiring an

Table 1. Participant Characteristics Across Pre-Intervention and Intervention Period.

| Variables | Total (n = 401) | | Pre-Intervention (n = 212) | | Intervention (n = 189) | |
|---|---|---|---|---|---|---|
| | N | % | N | % | N | % |
| **Age of the woman (years)** | | | | | | |
| <=24 | 35 | 8.7 | 22 | 10.4 | 13 | 6.9 |
| 25-35 | 270 | 67.3 | 141 | 66.5 | 129 | 68.2 |
| >35 | 96 | 23.9 | 49 | 23.1 | 47 | 24.9 |
| **Facility Name** | | | | | | |
| Kibagabaga DH | 122 | 30.4 | 50 | 23.6 | 72 | 38.1 |
| Nyamata DH | 78 | 19.5 | 58 | 27.4 | 20 | 10.6 |
| Nyanza DH | 116 | 28.9 | 50 | 23.6 | 66 | 34.9 |
| Ruhengeri RH | 85 | 21.2 | 54 | 25.4 | 31 | 16.4 |
| **Woman previously received an ultrasound examination at the hospital during this pregnancy** | | | | | | |
| No | 116 | 29.4 | 60 | 28.7 | 56 | 30.3 |
| Yes | 278 | 70.6 | 149 | 71.3 | 129 | 69.7 |
| **Status of conversation at first call** | | | | | | |
| Not yet reached the hospital | 24 | 12.7 | NA | NA | 24 | 12.7 |
| Reached hospital and delivered | 39 | 20.6 | NA | NA | 39 | 20.6 |
| Reached hospital and provided another appointment | 126 | 66.7 | NA | NA | 126 | 66.7 |
| **First contact at hospital after referral from health center at 36 + gestational age** | | | | | | |
| 36-38 Weeks | 189 | 52.4 | 71 | 37.6 | 118 | 68.6 |
| 39 + Weeks | 172 | 47.6 | 118 | 62.4 | 54 | 31.4 |
| **Uterine contraction at arrival** | | | | | | |
| No | 220 | 56.1 | 80 | 37.7 | 140 | 77.8 |
| Yes | 172 | 43.9 | 132 | 62.3 | 40 | 22.2 |

emergency cesarean delivery (CD) compared to those who did not have an ultrasound. Specifically, the odds of having an emergency CD were 73% lower for women who had a hospital ultrasound (0.27, [0.152; 0.456]).

## Discussion

The findings from the study highlight that during the intervention period a remarkable number of women reported that they had promptly presented to hospital within five days post-referral, as revealed within the first phone call conversation. Additionally, there was a considerable increase in the percentage of women who sought consultation within the optimal gestational age window of 36–38 weeks after referral from the health center, versus >39 weeks. This outcome is of particular significance as it indicates improved birth planning and proactive decision-making, which are associated with reduced risks of emergency CD. These observed results suggest that a well-coordinated non-clinical intervention can indeed make a significant contribution to improving timely access to health care services for pregnant women. These findings align with previous research, providing further support for the efficacy of non-clinical interventions in enhancing timely access to health care services for pregnant women. A study conducted in northern Ghana demonstrated similar positive findings, where the delivery of SMS/voice messages to pregnant women, educating them on maternal and child health and reminding them of their due dates for delivery, substantially increased skilled delivery rates at respective study sites compared to control sites [12]. Moreover, Martinez et al. observed improvement in linkages to higher-level obstetrical care with the use of mobile technology to support routine community-based practice in rural Guatemala [13].

**Table 2. Incidence of emergency CD per women's characteristics.**

| Variables | Total (n = 401) | Planned CD (n = 194) | Emergency CD (n = 207) | | | Chi-square P-value |
|---|---|---|---|---|---|---|
| | N | N | N | % | 95% C.I for % | |
| **Age of the woman** | | | | | | |
| <=24 years | 35 | 14 | 21 | 60 | [43.8; 76.2] | 0.1105 |
| 25-35 years | 270 | 125 | 145 | 53.7 | [47.8; 59.6] | |
| >35 years | 96 | 55 | 41 | 42.7 | [32.8; 52.6] | |
| **Facility Name** | | | | | | |
| Kibagabaga DH | 122 | 66 | 56 | 45.9 | [37.1; 54.7] | 0.026 |
| Nyamata DH | 78 | 26 | 52 | 66.7 | [56.2; 77.2] | |
| Nyanza DH | 116 | 60 | 56 | 48.3 | [39.2; 57.4] | |
| Ruhengeri RH | 85 | 42 | 43 | 50.6 | [40.0; 61.2] | |
| **Woman previously received an ultrasound examination at the hospital during this pregnancy** | | | | | | |
| No | 116 | 36 | 80 | 69 | [60.6; 77.4] | < 0.001 |
| Yes | 278 | 156 | 122 | 43.9 | [38.1; 49.7] | |
| **Gestational age at delivery** | | | | | | |
| Preterm (≤ 36 weeks) | 23 | 3 | 20 | 87 | [73.3; 100.0] | <0.001 |
| Term (37–38 weeks) | 94 | 41 | 53 | 56.4 | [46.4; 66.4] | |
| Full-term+ (>39 weeks) | 234 | 132 | 102 | 0 | [37.2; 49.9] | |
| **Uterine contraction at arrival** | | | | | | |
| No | 220 | 173 | 47 | 21.4 | [16.0; 26.8] | < 0.001 |
| Yes | 172 | 17 | 155 | 90.1 | [85.6; 94.6] | |
| **Period** | | | | | | |
| Pre-intervention | 212 | 65 | 147 | 69.3 | [63.1; 75.5] | < 0.001 |
| Intervention | 189 | 129 | 60 | 31.7 | [25.1; 38.3] | |

The reduction of emergency CD was further demonstrated by the logistic regression analysis as the odds of emergency CD during the intervention period were approximately 81% lower than during the pre-intervention period. This substantial reduction in emergency CD rates underscores the effectiveness of the active follow-up intervention in promoting better managed deliveries for pregnant women with previous CD in Rwanda. Hospital-based ultrasound examinations emerged as a key factor in reducing the likelihood of emergency CD. Women who underwent an ultrasound during pregnancy had substantially lower odds of requiring an emergency procedure. McLaughlin et al. reported that an intervention facilitating pregnant women with a previous CD in Burundi to access appointments for delivery and a complementary ultrasound examination around the delivery period led to a reduction in urgent repeat CDs [14].

The observed reduction of emergency CD can be attributed to tailored education provided to the women and the personalized phone call conversations aimed at continuing education within the community and reminding women to visit the hospital after referral. Through the educational component of the intervention, expectant mothers gained awareness of the benefits of discussing their delivery options and creating a birth plan with their health care providers. This increased understanding may have motivated them to proactively seek delivery planning at the hospital, resulting in better birth preparedness. Lund et al. noted that educational messages directed toward pregnant women offered support in various aspects, including comprehension of pregnancy danger signs, and fostering a sense of support from the health system. As a result, this influenced pregnant women's likelihood of attending health facilities [15].

Furthermore, personalized phone call conversations presented an opportunity for ongoing education post-referral. However, they did not serve as reminders for women to attend hospital appointments, as the majority had already done so before the conversation. The expectation of receiving a reminder call potentially heightened women's sense

**Table 3. Logistic regression: Factors associated with emergency caesarean delivery.**

| Variable | Unadjusted Model | | | Adjusted Model | | |
|---|---|---|---|---|---|---|
| | Odds Ratio | 95% C.I | P-value | Odds Ratio | 95% C.I | P-value |
| **Period** | | | | | | |
| Pre-intervention (ref) | | | | | | |
| Intervention | 0.206 | [0.135; 0.314] | < 0.001 | 0.187 | [0.115;0.298] | **<0.001** |
| **Age of the woman (years)** | | | | | | |
| <=24 (ref) | | | | | | |
| 25-35 | 0.773 | [0.377; 1.585] | 0.482 | 1.134 | [0.468; 2.692] | 0.777 |
| >35 | 0.497 | [0.226; 1.093] | 0.082 | 0.740 | [0.286; 1.878] | 0.528 |
| **Facility Name** | | | | | | |
| Kibagabaga DH (ref) | | | | | | |
| Nyamata DH | 2.357 | [1.306; 4.253] | 0.004 | 1.743 | [0.868; 3.542] | 0.121 |
| Nyanza DH | 1.100 | [0.661; 1.831] | 0.714 | 1.422 | [0.786; 2.594] | 0.247 |
| Ruhengeri RH | 1.207 | [0.693; 2.101] | 0.507 | 1.354 | [0.698; 2.635] | 0.370 |
| **Woman previously received an ultrasound examination at the hospital during this pregnancy** | | | | | | |
| No (ref) | | | | | | |
| Yes | 0.352 | [0.222; 0.557] | < 0.001 | 0.266 | [0.152; 0.456] | **<0.001** |

of responsibility toward their own health care. It is plausible that this anticipation instilled a heightened level of personal accountability for prenatal care, prompting proactive engagement without waiting to be reminded by health care providers. However, this suggests the necessity of a qualitative study to explore the underlying factors influencing women's health care-seeking behavior and their attitudes toward reminder calls, providing valuable insights for improving intervention strategies and enhancing maternal health care delivery.

These findings have important implications for policymakers, health care providers, and researchers. They underscore the need for targeted non-clinical interventions to mitigate adverse outcomes related to CDs and ensure safer births for women with higher risk pregnancies. By integrating evidence-based strategies into routine ANC, policymakers and health care providers can promote health care-seeking behavior among pregnant women and improve access to safe and quality obstetric and surgical care. By adopting comprehensive approaches that address both clinical and non-clinical aspects, progress toward safer and more successful deliveries for women with risky pregnancies can be achieved.

We acknowledge limitations in this study. The study did not analyze the contribution of each aspect of the intervention, and the analysis did not include maternal and newborn outcomes because it was out of the scope of the study as the primary objective was to evaluate the effectiveness of the intervention on the emergency CD rate among women with previous CD. In addition, the sample size of the study was too small to detect significant differences. Furthermore, the study lacks generalizability beyond the context of Rwanda's health system as the findings may be influenced by Rwanda's health care infrastructure, organization, and policies, which may not be representative of other countries' health systems. Another limitation of the study is the absence of data on key socioeconomic factors, such as education level and residence (urban/rural). The lack of these variables limits the ability to explore their potential influence on healthcare utilization and outcomes during the study period. Furthermore, the study did not account for external factors, such as the economic challenges caused by the COVID-19 pandemic. While the health facility characteristics remained consistent, financial constraints faced by some individuals may have delayed or prevented timely access to healthcare services. These unmeasured confounders could have influenced the study's findings. Future research should consider incorporating these variables to provide a more comprehensive understanding of the factors affecting healthcare access and outcomes.

Despite these limitations, the study highlights the importance of close and targeted follow-up of women with previous CD scars to reduce the rate of emergency CD among this population group.

## Conclusions

This study has demonstrated the effectiveness of an active follow-up intervention in promoting delivery planning and reducing emergency CD rates among pregnant women with previous CD scars in Rwanda. The comprehensive intervention including tailored education and personalized phone conversations around the delivery period appears to have contributed to increased awareness and motivation for women to seek timely care at the hospital for delivery planning. This is particularly important in settings where safe, monitored TOLAC is not widely available. Further research is needed to explore the long-term impacts and cost-effectiveness of these non-clinical interventions in various health care settings.

## Acknowledgments

The authors gratefully acknowledge the support of EngenderHealth and IntraHealth International for this work. Our acknowledgment also goes to the ANC nurses and respective health facilities for support during this study.

## Author contributions

**Conceptualization:** Josée Uwamariya, Gloriose Abayisenga, Ntwali Ndizeye, Fidele Nkurunziza, Christian Mazimpaka, Jean Maurice Munyabarenzi, Rosine Bigirimana, Dieudonne Ndatimana.

**Data curation:** Josée Uwamariya, Gloriose Abayisenga, Fidele Nkurunziza, Albert Bisore Ngemanyi, Gilbert Rukundo.

**Formal analysis:** Josée Uwamariya, Gilbert Rukundo.

**Methodology:** Josée Uwamariya, Gloriose Abayisenga, Ntwali Ndizeye, Fidele Nkurunziza, Albert Bisore Ngemanyi, Christian Mazimpaka, Jean Maurice Munyabarenzi, Rosine Bigirimana, Gilbert Rukundo, Dieudonne Ndatimana.

**Supervision:** Josée Uwamariya, Ntwali Ndizeye, Fidele Nkurunziza, Albert Bisore Ngemanyi, Rosine Bigirimana, Dieudonne Ndatimana.

**Validation:** Josée Uwamariya, Albert Bisore Ngemanyi, François Regis Cyiza, Farhad Khan, Karen Levin, Anne Fitzgerald Vinluan, Dieudonne Ndatimana.

**Visualization:** Josée Uwamariya, Gilbert Rukundo, Anne Fitzgerald Vinluan.

**Writing – original draft:** Josée Uwamariya, Ntwali Ndizeye, Anne Fitzgerald Vinluan, Kathryn Mimno.

**Writing – review & editing:** Josée Uwamariya, Ntwali Ndizeye, Christian Mazimpaka, Jean Maurice Munyabarenzi, Rosine Bigirimana, Gilbert Rukundo, François Regis Cyiza, Farhad Khan, Renae Stafford, Karen Levin, Anne Fitzgerald Vinluan, Kathryn Mimno, Dieudonne Ndatimana.

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
