## [Decision Letter · Decision Letter 0]

PONE-D-24-21352Effect of active follow-up of women with previous cesarean delivery on uptake of timely safe obstetric and surgical care: comparison between pre-intervention and intervention cohorts in RwandaPLOS ONE

Dear Dr. Josee,

Thank you for submitting your manuscript to PLOS ONE. After careful consideration, we feel that it has merit but does not fully meet PLOS ONE’s publication criteria as it currently stands. Therefore, we invite you to submit a revised version of the manuscript that addresses the points raised during the review process.

We look forward to receiving your revised manuscript.

Kind regards,

Orvalho Augusto, MD, MPH, PhD

Academic Editor

PLOS ONE

Journal Requirements:

2 We note that you have indicated that there are restrictions to data sharing for this study. PLOS only allows data to be available upon request if there are legal or ethical restrictions on sharing data publicly. For more information on unacceptable data access restrictions, please see http://journals.plos.org/plosone/s/data-availability#loc-unacceptable-data-access-restrictions .   

Additional Editor Comments:

This is an interesting work and an important study. Indeed, cesarean delivery (CD) prevalence is increasing or has increased in many parts of the globe. Interventions to screen and appropriately follow women with a previous CD is an imperative.

Here, the authors aim to assess the effectiveness of the "USAID MOMENTUM Safe Surgery in Family Planning and Obstetrics" intervention in reducing emergency cesarean delivery among women with previous CDs. They compare retrospectively two non-parallel cohorts of women delivering in four hospitals with a history of previous cesarean delivery (CD). One cohort includes these women between December 2021 and February 2022, and another includes women between November 2022 and May 2023. The authors conducted individual-level analysis and found a substantial "effectiveness."

Major issues:

1. Please be clear about the outcome and aim of this study:

- The authors indicate a single aim (I suggest removing the word primary in that case). Then, somewhere in the methods and results, the authors decide and embark in a quest of an additional aim to find factors associated with emergency CD. This is a serious confusion. Stick to the aim and may be to analyze the data for three outcomes (or 4): i) one as currently analysed emergency CD; ii) another is the point a) in line 98; and iii) another would be the point b) in line 99. A potential outcome would be the low birth weight which for reasons the authors need to explain they included this variable as predictor for CD [the issue here is you are using a cohort study and collect the birth weight after your outcome CD. This is highly problematic].

2. The authors did an analysis disregarding completely temporal confounding, seasonality, and inadequate strategy to select adjusting variables.

- Temporal confounding - the study happened during dynamic changes due to, for instance, COVID-19. Nowhere in the manuscript is a convincing argument that these four health facilities did not change in utilization and oher critical characteristics that may influence in the outcomes. It is imperative to show that the characteristics of these 4 health facilities remained virtually the same between these two cohorts. Without that we cannot attribute the changes we see to the intervention alone. For example, add in the results a i) description of the characteristics of the health facilities in each period and ii) at least discuss this.

Minor issues:

1. In the abstract, it needs to be clear that two cohorts (non-parallel) are being compared and what those cohorts are.

2. Introduction in line 87, please remove the word "primary" in light of the major issue.

3. In the setting description. Can you provide information as to whether these health facilities are rural or urban?

4. Please review your formula on line 197. It should be a minus on the denominator "P_1+P_2" i.e it should be "P_1 - P_2". See your reference 10.

5. We miss additional information about the women included in the study. At least education level and residence (urban/rural). The lack of these variables should be discussed.

6. Table 2:

- Change the titling of the table to "Incidence of ER per women's characteristics."

- Then keep i) absolute counts columns (three columns), ii) the proportion of emergency CD row percentage, a iii) 95% confidence interval for the proportion, and the current overall p-value.

- It would be interesting to repeat this table for pre-intervention and for post-intervention. This could be added as supplementary materials, but any substantive striking change should be commented on in the results.

7. Table 3, is the analysis of association using unadjusted and adjusted logistic regression models. Few issues:

- weight at birth seems to be a newborn outcome - It needs to be explained as to why is considered as an "exposure" here for emergency CD

- uterine contraction at arrival seems to be part of the definition of the outcome according to lines 98 and 99, so please do not adjust for this variable

- Potential seasonality in the outcome was not assessed. If there is seasonality, this needs to be addressed. Over the year, some months are not comparable to others

8. What variables were eliminated by the stepwise procedure? Currently, this is not the best procedure for finding a set of variables to adjust for.

9. As I suggested above, for additional outcomes, similar tables 2 and 3 should be built for each outcome and put in supplementary materials.

Reviewers' comments:

Reviewer's Responses to Questions

**Comments to the Author**

1. Is the manuscript technically sound, and do the data support the conclusions?

Reviewer #1: Yes

2. Has the statistical analysis been performed appropriately and rigorously?

Reviewer #1: Yes

3. Have the authors made all data underlying the findings in their manuscript fully available?

Reviewer #1: Yes

4. Is the manuscript presented in an intelligible fashion and written in standard English?

Reviewer #1: Yes

5. Review Comments to the Author

Reviewer #1: The study high lights an important intervention to improve post partum outcome as the follow up reduces un necessary cesarian sections and hence risk of post cesarean infections and other post partum complications.

6. PLOS authors have the option to publish the peer review history of their article (what does this mean? ). If published, this will include your full peer review and any attached files.

**Do you want your identity to be public for this peer review?** For information about this choice, including consent withdrawal, please see our Privacy Policy .

Reviewer #1: **Yes: ** Stephen Rulisa

---

## [Author Response · Author response to Decision Letter 1]

4 Feb 2025

A. RESPONSE TO MAJOR ISSUES

Response to comment 1:

• Thank you for your feedback regarding the clarity of the study's objective. We have revised the objective to ensure it accurately reflects the scope of the study and aligns with the analyses conducted. The updated objective is now twofold: “The objective of this study is twofold: (1) to evaluate the effect of the intervention on the emergency CD rate among women with previous cesarean delivery, and (2) to identify factors associated with emergency cesarean deliveries.” This revised objective addresses the concern about ambiguity in the study’s aim and is clearly stated in the manuscript (see lines 91–92). It clarifies the dual focus of the study and ensures consistency across the methods, and results sections.

• We appreciate your observation regarding the outcomes analyzed in the study. We would like to clarify that this study has a single outcome: the rate of emergency cesarean deliveries, as mentioned in line 101 of the revised manuscript. The text in lines 102–107 provides a detailed explanation of how the outcome of emergency cesarean delivery was defined. Specifically, we outlined two scenarios under which a cesarean delivery was classified as an emergency: a) a woman with a previous CD was admitted with contractions and a physician’s documented the need for emergency intervention, or b) a woman with a previous CD had a delivery plan and presented to the hospital on the scheduled date and was admitted without contractions; however, while awaiting the procedure, she began experiencing contractions and the physician confirmed the need for emergency intervention. To ensure further clarity and address potential ambiguity, we revised the manuscript to explicitly state that these criteria were applied consistently across both the pre-intervention and intervention cohorts to ensure uniformity in outcome classification. This addition is reflected in line 108-109.

• We appreciate your comment regarding the inclusion of low birth weight as a predictor for emergency cesarean delivery. Upon careful review, we recognize that including low birth weight as a predictor in the original manuscript was inappropriate, as it violates the temporal sequence between cause and effect. Specifically, birth weight is determined after the occurrence of the outcome (emergency cesarean delivery), which makes its use as a predictor in a cohort study design methodologically incorrect. In response to this concern, we have decided to remove low birth weight from the analysis.

Response to comment 2:

• We appreciate the reviewer’s observation regarding potential temporal confounding. During the study period, the four health facilities (Kibagabaga, Nyamata, Nyanza, and Ruhengeri) remained consistent in terms of staffing levels, number of deliveries and CD rate as outlined in the setting section (see line 119-145). However, we recognize that the economic impact of the COVID-19 pandemic, which disrupted livelihoods globally, could have influenced the ability of some individuals to access healthcare services in a timely manner. Unfortunately, our study did not collect data directly related to the socioeconomic status of participants or their financial challenges during this period, which limits our ability to directly assess this potential confounding factor. We have mentioned this as a limitation in the manuscript (line 364-370)

B. RESPONSE TO MINOR ISSUES:

Response to comment 1. We have revised the Methods section of the abstract to state that two non-parallel cohorts were compared. These changes can be found in the manuscript on lines 37–39.

Response to comment 2: We have removed the word "primary" and clarified the study’s objective for better clarity. These revisions can be found in the manuscript on lines 90–94

Response to comment 3: we would like to clarify that the health facilities included in the study are located in urban areas, but they function as referral centers for surrounding health centers situated in both urban and rural areas. We have updated the manuscript to reflect this information (see lines 115–117.

Response to comment 4: We have reviewed the formula and corrected the denominator, changing the “+” sign to a “-” sign, in accordance with the reference (Line 232)

Response to comment 5: Thank you for highlighting the importance of including additional information about the women in the study, such as education level and residence (urban/rural). Unfortunately, these variables were not collected because the data were extracted from antenatal care (ANC) registers, electronic medical records, delivery registers, and patients' files, and these sources do not contain information on education level or residency status (urban/rural).

We agree that the absence of these variables is a limitation of our study, as it restricts our ability to assess the potential influence of these factors on healthcare utilization and outcomes. This limitation has been acknowledged and discussed in the manuscript (see lines 361-364).

Response to comment 6:

• The title of Table 2 has been updated to "Incidence of Emergency CD by Women's Characteristics”.

• Table 2 has been revised to include the absolute counts, the proportion of emergency CD (row percentages), the 95% confidence intervals for the proportions, and the overall p-value, as suggested.

• Additionally, the same table has been generated separately for the pre-intervention and intervention periods. These tables have been included as supplementary materials to provide a more detailed comparison between the two periods.

Response to Comment 7: We have addressed this comment as follows:

• Weight at birth as an exposure: We acknowledge your observation that weight at birth is typically considered a newborn outcome. Based on your feedback, this variable has been removed from the analysis in Table 2.

• Uterine contraction at arrival: We agree that uterine contraction at arrival is part of the definition of the outcome. Therefore, this variable has been excluded from the adjusted logistic regression models, and Table 3 has been updated accordingly.

• Potential seasonality confounder: To address the potential influence of seasonality, we revised the facility characteristics described in the setting section to include the average monthly delivery rate for both the pre-intervention and intervention periods. (Starting line 125). The data showed that these averages were nearly identical across the two periods, suggesting no significant seasonal variation in the outcome. (line 121-145)

Response to comment 8: Upon review, we acknowledge that the mention of the stepwise method in the methodology section was a mistake. In fact, the stepwise procedure was not used in the selection of variables for the analysis. Given the limited number of variables in our dataset, there was no need to reduce the variable set further. Instead, we used bivariate and multivariable logistic regression models as described in the methodology section (line 204-208). We have revised the manuscript to remove any reference to the stepwise procedure.

Response to comment 9: As clarified in response to Comment 1, the study focuses on a single outcome: the rate of emergency cesarean deliveries, as stated in line 101 of the revised manuscript. Therefore, there is no need to build additional tables for other outcomes.

---

## [Decision Letter · Decision Letter 1]

Effect of active follow-up of women with previous cesarean delivery on uptake of timely safe obstetric and surgical care: comparison between pre-intervention and intervention cohorts in Rwanda

PONE-D-24-21352R1

Dear Dr. Josee,

We’re pleased to inform you that your manuscript has been judged scientifically suitable for publication and will be formally accepted for publication once it meets all outstanding technical requirements.

Kind regards,

Patrick Ifeanyi Okonta, MBBCh, MPH, FWACS, FMCOG, MD, DRH

Academic Editor

PLOS ONE

Additional Editor Comments (optional):

Reviewers' comments:

Reviewer's Responses to Questions

**Comments to the Author**

1. If the authors have adequately addressed your comments raised in a previous round of review and you feel that this manuscript is now acceptable for publication, you may indicate that here to bypass the “Comments to the Author” section, enter your conflict of interest statement in the “Confidential to Editor” section, and submit your "Accept" recommendation.

Reviewer #1: All comments have been addressed

2. Is the manuscript technically sound, and do the data support the conclusions?

Reviewer #1: Yes

3. Has the statistical analysis been performed appropriately and rigorously? 

Reviewer #1: Yes

4. Have the authors made all data underlying the findings in their manuscript fully available?

Reviewer #1: Yes

5. Is the manuscript presented in an intelligible fashion and written in standard English?

Reviewer #1: Yes

6. Review Comments to the Author

Reviewer #1: The comments have been addressed and manuscript :Effect of active follow-up of women with previous cesarean delivery on uptake of timely

safe obstetric and surgical care: comparison between pre-intervention and intervention

cohorts in Rwanda is of good importance to health care system.

the objectives are in line with the title and address the challenges.

7. PLOS authors have the option to publish the peer review history of their article (what does this mean? ). If published, this will include your full peer review and any attached files.

**Do you want your identity to be public for this peer review?** For information about this choice, including consent withdrawal, please see our Privacy Policy .

Reviewer #1: **Yes: ** Stephen Rulisa

---

## [Editor Report · Acceptance letter]

PONE-D-24-21352R1

PLOS ONE

Dear Dr. Uwamariya,

I'm pleased to inform you that your manuscript has been deemed suitable for publication in PLOS ONE. Congratulations! Your manuscript is now being handed over to our production team.

Kind regards,

on behalf of

Professor Patrick Ifeanyi Okonta

Academic Editor

PLOS ONE